# Epipolar Prompt: A Simple Baseline for Motion Segmentation

## Abstract

Reconstructing dynamic 3D/4D scenes from uncalibrated videos remains challenging due to moving objects violating the static assumptions required for multi-view consistency. While motion segmentation can resolve this, existing methods struggle to generalize across datasets and ignore 3D geometry cues. To this end, we propose *Epipolar Prompt*, a zero-shot framework that synergizes epipolar geometry with foundation segmentation models (e.g., SAM) to achieve robust motion segmentation. Our approach first computes epipolar error maps from optical flow correspondences to localize regions that violate static scene assumptions. These error maps then guide an iterative prompt selection strategy to generate precise segmentation from SAM. Surprisingly, our simple yet effective prompt-based method outperforms both supervised and unsupervised approaches on standard benchmarks (e.g., +9.3 IoU over DAVIS2017) and demonstrates strong generalization to in-the-wild videos. Furthermore, we show that our motion masks serve as a plug-and-play enhancement for existing dynamic 4D reconstruction methods, leading to improved performance. View results at: https://anonymous-for.github.io/ICLR-4426/

## 1 Introduction

Motion segmentation (MS) aims to separate moving objects from static background in a video. Unlike traditional video object segmentation (VOS) that segments objects based on appearance, motion segmentation only segments objects undergoing independent 3D motion relative to the world coordinate system. This capability is essential for various applications, including uncalibrated 4D reconstruction (Wang et al., 2024b;a; Zhang et al., 2024; Lei et al., 2024; Liu et al., 2023; Zhang et al., 2022; Li et al., 2024), autonomous driving (Klappstein et al., 2009; Rashed et al., 2019), SLAM (Wang et al., 2020; Bescos et al., 2018), and augmented reality (AR) (Hammer et al., 2016).

Despite being conceptually simple, however, existing motion segmentation methods **struggle** to achieve both **high accuracy** and **strong generalization**. *Unsupervised methods* require no labeled data but often fall short in accuracy compared to supervised approaches. *Supervised methods* achieve high accuracy on specific datasets but suffer from poor generalization. These dilemmas are primarily due to two key issues: **absence of geometric cues** and **lack of data**.

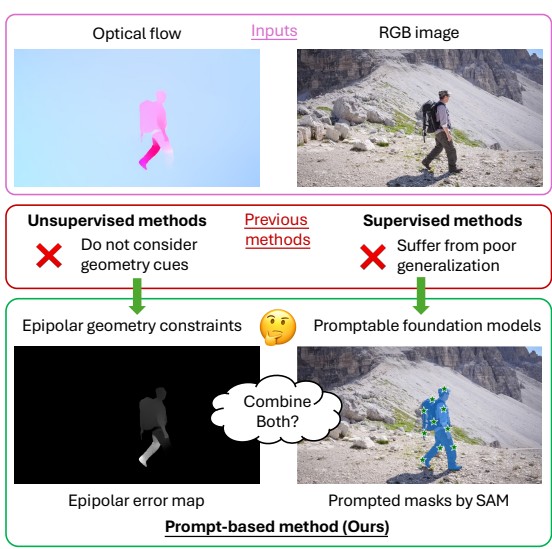

Figure 1: Motion segmentation often relies on optical flow and RGB images as inputs. However, existing unsupervised and supervised methods struggle to balance accuracy and generalization. In contrast, we introduce a novel prompt-based approach that leverages geometric cues and visual foundation models in a zero-shot manner, offering a new insight on motion segmentation.

**Geometric Cues.** Most existing motion segmentation (MS) methods attempt to identify moving regions from optical flow using clustering or learning-based approaches. However, in a truly static scene, moving regions inherently violate *multi-view geometric constraints* (Schonberger & Frahm, 2016). Yet, existing methods fail to incorporate these geometric cues, making it difficult to distinguish between apparent 2D motion and true motion in world coordinates.

**Lack of data.** MS requires true and complete motion labels, which are costlier to obtain than conventional VOS annotations. Models trained on a particular dataset often suffer from poor generalization due to the limited availability of real-world annotations. These models tend to overfit to their training data, making them unreliable in new scenarios or with new types of moving objects.

For example, even the most advanced motion segmentation models struggle to generalize effectively. The state-of-the-art (SOTA) FlowP-SAM (Xie et al., 2024b) is a supervised approach that releases separate model weights trained on DAVIS 2016 (Perazzi et al., 2016a) and DAVIS 2017 (Pont-Tuset et al., 2017). As shown in Table 1, when the model trained on DAVIS 2016 is tested on DAVIS 2017, its performance drops substantially compared to the version trained directly on DAVIS 2017. This highlights the fundamental challenge of generalizing across diverse real-world scenarios.

| $IoU \uparrow$ | FlowP-SAM (17) | FlowP-SAM (16) |
|---|---|---|
| DAVIS17 | 68.6 | 49.6 (19.0 ↓) |

Table 1: Performance evaluation of FlowP-SAM on DAVIS 2017 using the official implementation. Here, FlowP-SAM (17) refers to the model trained on DAVIS 2017, and FlowP-SAM (16) denotes the model trained on DAVIS 2016. Despite the similarity between these two datasets, the significant performance gap highlights the model's poor generalization ability. A similar issue has also been reported in their GitHub discussions [1].

To address these limitations, we propose *Epipolar Prompt*, a simple yet effective prompt-based motion segmentation method that integrates epipolar geometry with foundation models like Segment Anything (SAM) (Kirillov et al., 2023), enabling accurate segmentation in a zero-shot manner.

At its core, Epipolar Prompt leverages epipolar geometry to guide SAM's segmentation. First, we compute the epipolar error map (Longuet-Higgins, 1981) from video frames using 2D correspondences derived from optical flow. This error map highlights regions that violate the assumptions of rigid camera motion. We then sample points from high-error regions and use them as prompts for SAM, which segments the moving objects. By integrating epipolar geometry with SAM's strong generalization ability, our method enhances generalization across diverse scenes.

While intuitive, automating prompt selection requires balancing three key criteria. First, the resulting masks should align with strong *geometric cues* (i.e., high epipolar error). Second, the segmentation should be *semantically complete*, capturing entire objects rather than fragmented regions. Finally, *efficiency*, as selecting too many points can lead to excessive computational costs.

To achieve this balance, we propose a dedicated prompt selection scheme, which iteratively applies farthest point sampling (FPS) to generate well-distributed point prompts. In each iteration, the resulting masks are evaluated using a combination of confidence score, stability score, and epipolar adherence. In addition, we employ heuristic filtering to discard implausible segmentations, thereby improving the overall reliability of the segmentation.

By integrating these techniques, our final segmentation results are both highly accurate and robust across a variety of scenarios and datasets, demonstrating SOTA performance. On the challenging multi-object motion segmentation benchmark DAVIS 2017, we obtain a 9.3 improvement in IoU. This robust segmentation can be directly applied to 4D reconstruction, enhancing existing methods.

To summarize, our main contributions are as follows:

- A novel paradigm that formulates motion segmentation as a point prompt selection problem, providing a simple, fast, and training-free solution.

- An epipolar-guided point prompt sampling strategy to iteratively derive motion masks, and a suite of heuristic filtering techniques to effectively suppress false positive masks.

- State-of-the-art motion segmentation results on single-object and multi-object benchmarks, surpassing both unsupervised and supervised methods.

---

[1] https://github.com/Jyxarthur/flowsam/issues/6

- We further demonstrate that our motion masks effectively enhance 4D reconstruction in dynamic scenes, highlighting the practical significance of our method.

## 2 RELATED WORKS

### 2.1 VIDEO OBJECT SEGMENTATION

Video Object Segmentation (VOS) aims to produce dense masks for one of multiple "foreground" objects in videos. It can be categorized into two primary protocols, as outlined by several benchmark datasets (Caelles et al., 2019; Perazzi et al., 2016b): semi-supervised VOS (Caron et al., 2021; Jabri et al., 2020; Lai et al., 2020; Lai & Xie, 2019; Miao et al., 2022; Wang et al., 2019) and unsupervised VOS (Cho et al., 2019; Li et al., 2018; Lu et al., 2019; Luiten et al., 2020; Ventura et al., 2019). The semi-supervised approach needs to provide first-frame GT masks and aims to generate segmentation masks for subsequent frames. In contrast, unsupervised VOS automatically identifies and tracks salient objects throughout the video, without requiring any initial annotations.

### 2.2 MOTION SEGMENTATION

Unlike video object segmentation (VOS), motion segmentation (MS) seeks to automatically identify and segment *objects that are moving in world coordinates*, even under camera motion. However, high-quality datasets specifically annotated for motion segmentation are scarce. Most existing approaches rely on VOS datasets, but as shown in Figure 2, the labeled objects do not always include all truly moving objects. This mismatch leads many supervised MS methods trained on such data to behave more like salient object segmentation methods rather than genuine motion segmentation.

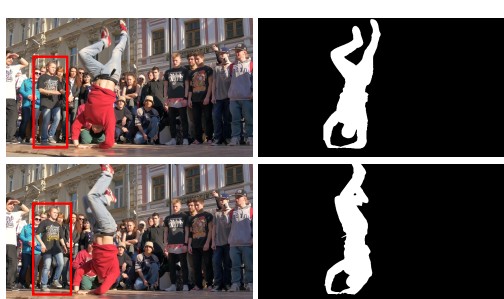

Figure 2: Sample images and annotations from DAVIS 2016 (Perazzi et al., 2016a). Only the most salient person is annotated, while the person (in red box) is moving but remains unlabeled.

Prior work has often focused on clustering pixels based on motion patterns (Keuper et al., 2015; Ochs & Brox, 2011; Brox & Malik, 2010; Xie et al., 2019) or applying motion models to optical flows to predict segmentations (Bideau & Learned-Miller, 2016; Bideau et al., 2018; Mahendran et al., 2018; Meunier et al., 2022). Advances in model architectures (Dave et al., 2019; Tokmakov et al., 2019; Yang et al., 2021a) and training strategies (Yang et al., 2019; Lamdouar et al., 2021; Xie et al., 2022) have significantly improved segmentation accuracy. Recently, with the advancement of vision foundation models, several methods (Xie et al., 2024b; Goli et al., 2024; Karazija et al., 2024; Huang et al., 2025) have explored using SAM (Kirillov et al., 2023; Ravi et al., 2024) or trackers (Doersch et al., 2024; Karaev et al., 2024) in their training or optimization processes to enhance motion segmentation. Our approach also uses SAM but focuses on sampling point prompts rather than relying on gradient descent.

## 3 METHOD

For a video consisting of $T$ frames, we denote the image sequences as $\{I_t\}$ where $t \in \{1, ..., T\}$, the aim of motion segmentation is to predict binary masks $\{\mathcal{M}_t\}$ indicating all pixels belonging to moving objects. Our method first uses epipolar geometry constraints to identify areas that have motion relative to ego-motion (Sec. 3.1, Figure 3). Since these areas usually are semantically incomplete, an iterative point sampling strategy is used to prompt SAM to generate masks (Sec. 3.2, Figure 4). Lastly, we filter the false positives by utilizing heuristic knowledge (Sec. 3.3, Figure 6). Notably, our approach *does not require training, fine-tuning, or test-time optimization*, making it simple, fast, and highly generalizable to in-the-wild scenes.

## 3.1 EPIPOLAR ERROR MASK DERIVATION

Epipolar geometry (Longuet-Higgins, 1981) characterizes the geometric relationship between two views of the same 3D scene. It imposes constraints on how points in one image correspond to points in the other, assuming the points belong to the static scene. In a purely rigid scene with known camera motion, every point correspondence should satisfy the epipolar constraint. However, when an object moves independently of the camera, its corresponding points deviate from this constraint. This deviation, measured by epipolar error, enables the detection of moving objects within the scene.

**Correspondence Estimation.** We start with predicting optical flow $F_{t \to t'}$ between two frames $I_t$ and $I_{t'}$ using an off-the-shelf model, RAFT (Teed & Deng, 2020). Optical flow provides a dense field of pixel correspondences, representing per-pixel displacement vectors between the frames. Let $\mathbf{x}_t = (x, y, 1)^T$ be a homogeneous coordinate in frame $I_t$, and let its corresponding point in $I_{t'}$, determined by optical flow, be $\mathbf{x}_{t'}$. The displacement is denoted as: $\mathbf{x}_{t'} = \mathbf{x}_t + \mathbf{d}$, where $\mathbf{d} = (u, v, 0)^T$ is the optical flow vector.

**Fundamental Matrix.** To leverage epipolar geometry, we estimate the fundamental matrix $\mathbf{F} \in \mathbb{R}^{3 \times 3}$, which encodes the epipolar constraints between the two frames. The fundamental matrix satisfies the epipolar constraint for any corresponding points:

$$\mathbf{x}_{t+1}^T \mathbf{F} \mathbf{x}_t = 0 \qquad (1)$$

The fundamental matrix is essential because it encapsulates the intrinsic geometric relationship between two views without requiring knowledge of the camera parameters. In general, $\mathbf{F}$ can be estimated using the 7-point or 8-point algorithm. However, since the scene contains dynamic objects, which act as outliers, a robust estimation method is necessary.

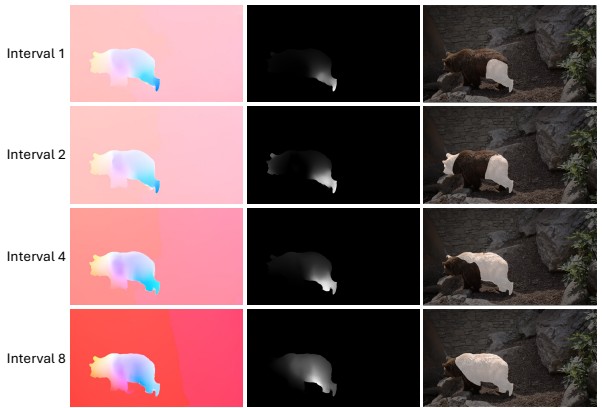

Figure 3: Visualization of the derived epipolar error masks across different time intervals. From left to right: optical flow, epipolar error map, and epipolar error mask.

We employ the Least Median of Squares (LMedS) method (Rousseeuw, 1984) to robustly estimate the fundamental matrix. It works by randomly selecting eight point correspondences and computing a candidate fundamental matrix. The algorithm repeats this process multiple times and selects the matrix that results in the lowest median error:

$$\mathbf{F} = \arg\min_{\mathbf{F}} \quad \text{median} \left( \left( \mathbf{x}_{t'}^T \mathbf{F} \mathbf{x}_t \right)^2 \right) \qquad (2)$$

Compared to classic RANSAC (FISCHLER AND, 1981) method, LMedS does not require setting an inlier threshold, making it particularly effective in cases where the noise distribution is unknown or highly variable. And we empirically find that LMedS provides superior robustness.

**Epipolar Error.** Once $\mathbf{F}$ is obtained, we compute the epipolar error map to quantify the deviation of correspondences from the epipolar constraint. Since the direct algebraic error is known to be sensitive to noise and does not accurately reflect geometric error (Longuet-Higgins, 1981), we use the Sampson error (Hartley & Zisserman, 2003) instead, which provides a first-order approximation of the geometric reprojection error:

$$E_{t \to t'}(\mathbf{x}_t) = \frac{\left( \mathbf{x}_{t'}^T \mathbf{F} \mathbf{x}_t \right)^2}{(\mathbf{F}\mathbf{x}_t)_1^2 + (\mathbf{F}\mathbf{x}_t)_2^2 + (\mathbf{F}^T\mathbf{x}_{t'})_1^2 + (\mathbf{F}^T\mathbf{x}_{t'})_2^2} \qquad (3)$$

where $(\mathbf{F}\mathbf{x}_t)_1$ and $(\mathbf{F}\mathbf{x}_t)_2$ denote the first and second components of the epipolar line equation. The Sampson error corrects for scale variations and significantly improves robustness in detecting outlier correspondences. By thresholding the error map, we obtain a epipolar error mask $M_{t \to t'}$ which indicates the moving areas determined by the epipolar geometry constraints:

$$M_{t \to t'} = \begin{cases} 1, & \text{if } E_{t \to t'} > \tau, \\ 0, & \text{otherwise,} \end{cases} \qquad (4)$$

Where $\tau$ is a threshold that indicates when the motion is significant enough to be categorized as a moving area. An illustration of the epipolar error masks is shown in Figure 3.

## 3.2 EPIPOLAR-GUIDED POINT PROMPT SAMPLING

While the epipolar error mask $M_{t \to t'}$ provides valuable hints for moving areas, it is usually incomplete in semantics. To address this, we leverage the Segment Anything Model (SAM) (Xie et al., 2022), a promptable vision foundation model capable of generating semantic segmentation masks given point prompts. A straightforward approach is to sample grid points within the epipolar error mask region and use them as prompts. However, a high grid density increases computational cost, while a low density risks missing some moving regions.

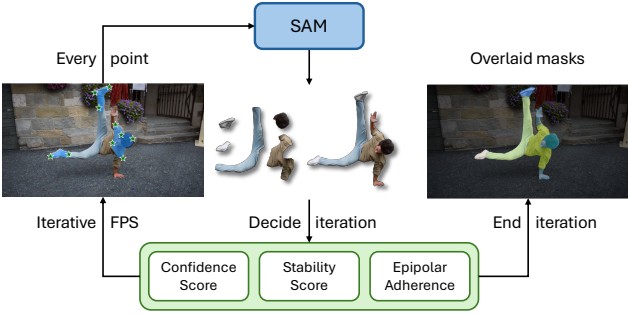

Figure 4: Epipolar-guided point prompt sampling. At each iteration, points are sampled using FPS within the epipolar error mask (highlighted in blue). Each point is used to prompt SAM to generate masks. The iteration continues until the masks satisfy all three conditions.

To this end, we propose an epipolar-guided sampling method that achieves optimization objectives by iterative sampling operations, as shown in Figure 4. The objectives are: *Adherence to epipolar cues* - The resulting masks should cover all regions with high epipolar error and avoid non-epipolar areas; *Semantic completeness* - The motion mask should capture the full extent of moving objects; *Efficiency* - The method should minimize the number of prompts while maintaining high accuracy.

To achieve these objectives, we employ the farthest point sampling (FPS) technique (Qi et al., 2017) iteratively. In each iteration, we sample $N$ points from the epipolar error mask using FPS, ensuring well-distributed coverage while avoiding the redundancy of grid-based sampling. Each sampled point $P_i$ is used to prompt SAM, generating a candidate mask $M_{SAM}$.

$$\{M_{SAM}\} = \{\text{SAM}(P_i) \mid P_i \in \text{FPS}(M_{t \to t'})\} \tag{5}$$

Since SAM's predictions are not always reliable, we apply the same *confidence* and *stability* score filtering as in SAM (Kirillov et al., 2023). Additionally, we discard invalid masks that do not overlap with the epipolar error mask, to ensure epipolar adherence.

Then, by taking the union of all valid masks, we obtain the motion mask $\mathcal{M}_t = \bigcup \{M_{SAM}\}$. The iteration ends when the motion mask $\mathcal{M}_t$ covers all regions in the epipolar error mask ($\mathcal{M}_t \cap M_{t \to t'} = M_{t \to t'}$) or no new valid masks are generated, which usually takes a few iterations.

This simple approach ensures efficiency through FPS, semantic completeness through SAM, and adher-

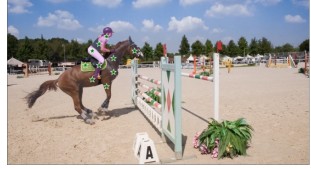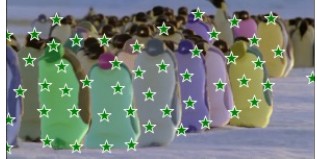

Figure 5: Examples of point sampling requiring 1 and 5 iterations, with each iteration selecting 10 points via FPS. The iterative strategy adaptively adjusts point prompts to scenes with varying numbers, shapes, and sizes of moving objects.

ence to epipolar cues through iterative refinement. It is effective regardless of the number, shape, or size of regions in the epipolar error mask, balancing between accuracy and computational cost, as shown in Figure 5.

Moreover, to enhance robustness, the sampling is conducted on aggregated epipolar error maps from optical flow estimates over multiple time intervals in both forward and backward directions:

$$t' \in \{t \pm 2^n \mid n = 0, 1, 2, 3\} \tag{6}$$

This multi-scale temporal consideration effectively captures both subtle and large motions, improving the completeness of the motion segmentation.

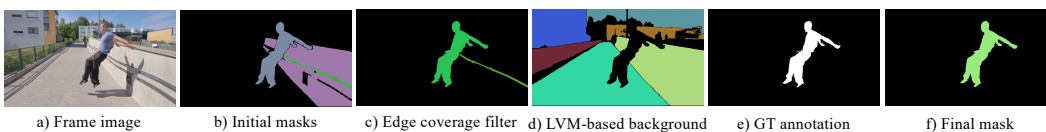

a) Frame image     b) Initial masks     c) Edge coverage filter     d) LVM-based background     e) GT annotation     f) Final mask

Figure 6: Illustration of heuristic filter of false positives. **a)** show the frame image and , respectively. **b)** shows the initial masks obtained before heuristic filtering. **c)** demonstrates edge coverage filtering, which removes a background mask. **d)** displays the VLM-based background masks, which is more comprehensive. Finally, **e)** shows the final mask after eliminating VLM-based backgrounds. **f)** show the ground truth annotation.

### 3.3 HEURISTIC FILTERING OF FALSE POSITIVE

The previous steps though simple, yet already produce high-quality results. However, rare false positives may still occur, often due to factors such as blur or illumination changes that affect optical flow estimation. To handle these cases, we propose two heuristic approaches to filter out the false positives, inspired by the fact that humans can easily identify static regions even from a single image.

The first technique is **edge coverage filtering**, a rule-based method that is computationally efficient. Background regions such as the sky or ground often extend to multiple image edges and cover plenty of edge pixels, whereas a moving object is unlikely to do so, as this would imply an unrealistically large motion region, which contradicts the assumption that static areas dominate the scene. We define the *edge coverage* of a mask as the ratio of mask pixels that touch the image edges to the total number of pixels along those edges. A mask is considered background and discarded if its edge coverage exceeds $0.4$.

The second technique is **VLM-based filtering**, leveraging the strong visual question answering (VQA) ability of vision-language models (VLMs), such as Qwen-VL (Bai et al., 2023) to eliminate more background regions. We prompt Qwen-VL to identify elements in the image that are likely to be moving (e.g., people, vehicles) and those that are likely static (e.g., ground, sky). We then utilize Grounded-SAM (Ren et al., 2024), which generates segmentation masks from text prompts, to obtain segmentations for static and moving regions. The final background masks are determined by subtracting the moving object masks from the static ones. An example of these two approaches is shown in Figure 6. Note that this step is only an additional refinement to remove rare false positives (see Table 8 in the Supplement).

## 4 EXPERIMENTS

### 4.1 IMPLEMENTATION DETAILS

Our method requires neither training nor test-time optimization. All experiments are conducted on a single NVIDIA RTX A5000 GPU. To improve computational efficiency in fundamental matrix estimation, we apply grid sampling to the dense optical flow field instead of using all points. The grid size is set to 8. The epipolar error threshold $\tau$ is $0.08$. Additionally, the resulting epipolar error map is eroded by 5 pixels to avoid ambiguous boundaries. For generating prompts, we sample $N = 10$ points per iteration to ensure a sufficient number of prompts. The confidence score, stability score, and invalid mask IoU thresholds are set to 0.88, 0.95, and 0.01, respectively. For VLM-based background filtering, we employ QWen2.5-VL-7B. The text prompt and example results are provided in the supp.

### 4.2 MOTION SEGMENTATION

**Datasets and Evaluation.** Following prior works (Xie et al., 2022; 2024b;a), we evaluate our method on a multiple benchmarks, including single-object datasets such as DAVIS 2016 (Perazzi et al., 2016a), SegTrack v2 (Li et al., 2013), and FBMS-59 (Ochs et al., 2013), as well as multi-object datasets like DAVIS 2017 (Pont-Tuset et al., 2017) and YouTube-VOS2018-motion (Xie et al., 2024a; Xu et al., 2018). These datasets are sourced from real-world videos, encompassing diverse

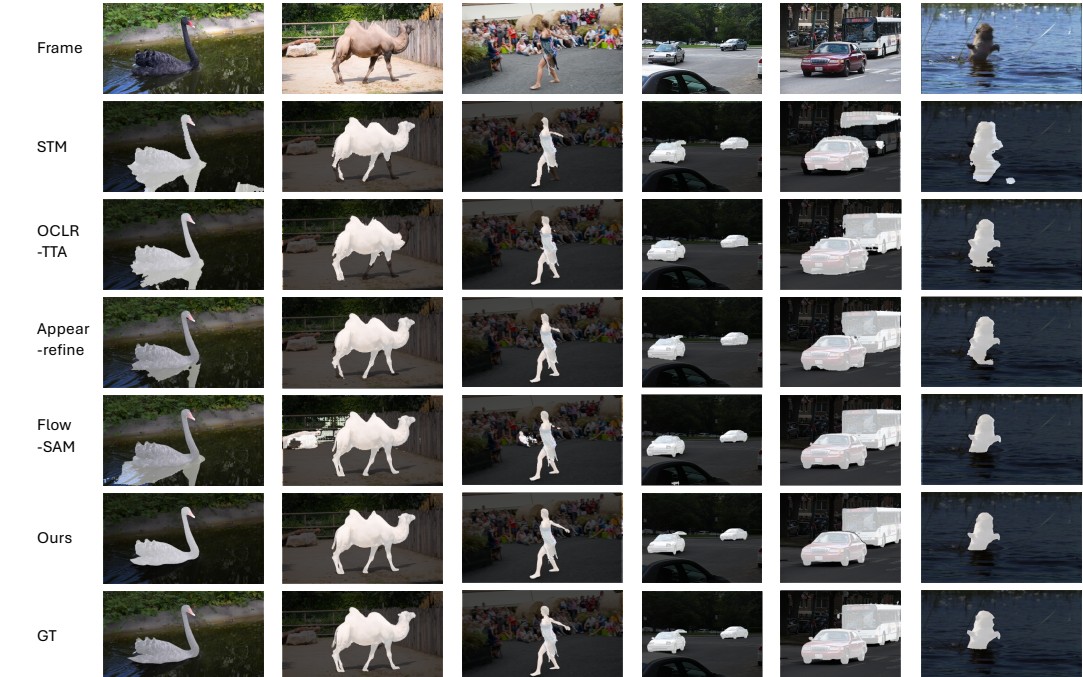

Figure 7: **Qualitative comparison** of motion segmentation among different methods.

object categories, background variations, and complex motion dynamics of both objects and the camera.

For DAVIS 2016, we evaluate on its validation set, which contains 20 videos. For SegTrack v2 and FBMS-59, we follow previous works (Lamdouar et al., 2021; Yang et al., 2021a) by grouping all moving objects into a single mask when evaluating on their respective test sets, which consist of 14 and 30 videos. The DAVIS 2017 validation set includes 30 videos, each with distinct annotations of multiple predominantly moving objects. YouTube-VOS2018-motion (Xie et al., 2024a) consists of 120 videos, curated from the original YouTube-VOS2018 (Xu et al., 2018) dataset by excluding sequences with common motion patterns, severe partial motion, or stationary objects.

It is important to note that these datasets do not provide exhaustive annotations for all moving objects, as they primarily focus on salient foreground objects (shown in Figure 2. Since our method segments all moving objects, we follow prior works (Yang et al., 2021a; Xie et al., 2022; 2024a;b) by applying Hungarian matching between predictions and annotations before evaluation. We report per-frame IoU (Intersection over Union) as our evaluation metric.

**Baselines.** To ensure a comprehensive evaluation, we compare our method against a diverse set of approaches. The unsupervised methods include COD (Lamdouar et al., 2020), Motion Group (Yang et al., 2021a), EM (Meunier et al., 2022), STM (Meunier & Bouthemy, 2023), Deformable Sprite (Ye et al., 2022), and DystaB (Yang et al., 2021b). The supervised methods include SIMO (Lamdouar et al., 2021), MATNet (Zhou et al., 2020), OCLR (Xie et al., 2022), Appear-refine (Xie et al., 2024a), and Flow-SAM (Xie et al., 2024b). We report frame-level evaluation results as presented in their respective papers. Notably, we classify methods trained on either synthetic (Lamdouar et al., 2021; Xie et al., 2022; 2024a) or human-labeled (Zhou et al., 2020; Xie et al., 2024b) datasets as supervised, given their reliance on data-driven learning.

**Quantitative Results.** Table 2 presents a performance comparison across both single-object and multi-object benchmarks, along with their respective input settings. Despite its simplicity, our zero-shot prompt-based method significantly outperforms previous unsupervised approaches (e.g., 8.2 IoU improvement on FBMS59). Furthermore, our method even achieves superior results compared to supervised methods (e.g., 9.3 IoU improvement on DAVIS 2017). It is worth noting that SegTrack v2 and FBMS are relatively old datasets, containing many low-quality or even corrupted frames (see

| | Input settings | | Multi-object benchmarks | | Single-object benchmarks | | |
|---|---|---|---|---|---|---|---|
| Methods | Flow | RGB | YTVOS18-m | DAVIS17 | DAVIS16 | SegTrack v2 | FBMS59 |
| *Unsupervised methods* | | | | | | | |
| COD (Lamdouar et al., 2020) | ✓ | ✗ | – | – | 65.3 | – | – |
| Motion Grouping (Yang et al., 2021a) | ✓ | ✗ | **37.0** | **38.4** | 68.3 | 58.6 | 53.1 |
| EM (Meunier et al., 2022) | ✓ | ✗ | – | – | 69.3 | 55.5 | 57.8 |
| STM (Meunier & Bouthemy, 2023) | ✓ | ✗ | – | – | 73.2 | 55.0 | 59.2 |
| Deformable Sprite (Ye et al., 2022) | ✓ | ✓ | – | – | 79.1 | 72.1 | 71.8 |
| DystaB (Yang et al., 2021b) | ✓ | ✓ | – | – | **82.8** | **74.2** | **75.8** |
| *Supervised methods* | | | | | | | |
| SIMO (Lamdouar et al., 2021) | ✓ | ✗ | – | – | 67.8 | 62.2 | – |
| MATNet (Zhou et al., 2020) | ✓ | ✓ | – | 56.7 | 82.4 | 50.4 | 76.1 |
| OCLR-flow (Xie et al., 2022) | ✓ | ✗ | 46.5 | 54.5 | 72.1 | 67.6 | 70.0 |
| OCLR-TTA (Xie et al., 2022) | ✓ | ✗ | 49.5 | 55.7 | 73.3 | 65.9 | 70.5 |
| RoMo (Goli et al., 2024) | ✓ | ✓ | – | – | 77.3 | 67.7 | 75.5 |
| FlowI-SAM (Xie et al., 2024b) | ✓ | ✗ | 58.6 | 65.7 | 79.1 | 70.1 | 75.1 |
| Appear-refine (Xie et al., 2024a) | ✓ | ✓ | 67.1 | 67.0 | 81.1 | 76.6 | 81.9 |
| FlowP-SAM (Xie et al., 2024b) | ✓ | ✓ | 76.9 | 69.9 | 86.1 | 83.9 | 87.9 |
| FlowP-SAM+FlowI-SAM | ✓ | ✓ | **77.4** | **71.6** | **86.2** | **84.2** | **88.7** |
| *Prompt-based method (training-free)* | | | | | | | |
| **Epipolar Prompt (Ours)** | ✓ | ✓ | **72.5** | **79.2** | **87.3** | **79.1** | **84.0** |

Table 2: **Quantitative comparison** on motion segmentation benchmarks, measured by $IoU \uparrow$.

| Datasets | Methods | ATE ↓ | RPEt ↓ | RPEr ↓ |
|---|---|---|---|---|
| Sintel | DUSt3R w/o mask | 0.4161 | 0.2168 | 18.0381 |
| | DUSt3R w/ MonST3R mask | 0.3357 | 0.1555 | 0.9464 |
| | DUSt3R w/ FlowP-SAM mask | 0.3791 | 0.1682 | 0.9679 |
| | **DUSt3R w/ our mask** | **0.3344** | **0.1549** | **0.9288** |
| | MonST3R w/o mask | 0.1577 | 0.0993 | 1.9241 |
| | MonST3R w/ MonST3R mask | 0.1093 | 0.0428 | 0.7369 |
| | MonST3R w/ FlowP-SAM mask | 0.1216 | 0.04645 | 0.8333 |
| | **MonST3R w/ our mask** | **0.1036** | **0.0400** | **0.7054** |
| Bonn | DUSt3R w/o mask | 0.0345 | 0.0159 | 1.6419 |
| | DUSt3R w/ MonST3R mask | 0.0268 | 0.0074 | 0.6064 |
| | DUSt3R w/ FlowP-SAM mask | 0.0311 | 0.0077 | 0.6442 |
| | **DUSt3R w/ our mask** | **0.0241** | **0.0073** | **0.5985** |
| | MonST3R w/o mask | 0.0346 | 0.0069 | 0.6913 |
| | MonST3R w/ MonST3R mask | 0.0274 | 0.0069 | 0.6153 |
| | MonST3R w/ FlowP-SAM mask | 0.0346 | 0.0069 | 0.6912 |
| | **MonST3R w/ our mask** | **0.0216** | **0.0067** | **0.5977** |

Table 3: **Camera pose evaluation** with different motion masks.

| Datasets | Methods | Abs Rel ↓ | RMSE ↓ | $\delta < 1.25$ ↑ |
|---|---|---|---|---|
| Sintel | DUSt3R w/o mask | 0.502 | 5.1411 | 0.5488 |
| | DUSt3R w/ MonST3R mask | 0.4820 | 5.0536 | 0.5638 |
| | DUSt3R w/ FlowP-SAM mask | 0.4790 | 5.1110 | 0.5698 |
| | **DUSt3R w/ our mask** | **0.4787** | **5.0526** | **0.5785** |
| | MonST3R w/o mask | 0.3334 | 4.5452 | 0.5910 |
| | MonST3R w/ MonST3R mask | 0.3335 | 4.5012 | 0.5887 |
| | MonST3R w/ FlowP-SAM mask | 0.3301 | 4.5361 | 0.5894 |
| | **MonST3R w/ our mask** | **0.3279** | **4.4453** | **0.6075** |
| Bonn | DUSt3R w/o mask | 0.1491 | 0.4221 | 0.8441 |
| | DUSt3R w/ MonST3R mask | 0.1438 | 0.4138 | 0.8444 |
| | DUSt3R w/ FlowP-SAM mask | 0.1466 | 0.4199 | 0.8439 |
| | **DUSt3R w/ our mask** | **0.1432** | **0.4120** | **0.8466** |
| | MonST3R w/o mask | 0.06723 | 0.2608 | 0.9568 |
| | MonST3R w/ MonST3R mask | 0.0625 | 0.2512 | 0.9624 |
| | MonST3R w/ FlowP-SAM mask | 0.0663 | 0.2579 | 0.9587 |
| | **MonST3R w/ our mask** | **0.0613** | **0.2494** | **0.9645** |

Table 4: **Depth estimation** with different motion masks.

Supp). As a result, our method performs slightly worse than the state-of-the-art supervised approach, FlowP-SAM, on these datasets. Nonetheless, it still outperforms all other methods.

**Qualitative Results.** As shown in Figure 7, our method generates more accurate and semantically complete segmentation results (columns 2 and 3). It effectively captures all moving objects in the video (columns 4 and 5) while avoiding background interference, even in challenging water scenarios (columns 1 and 6).

## 4.3 4D RECONSTRUCTION

Recent advances in 4D reconstruction (Wang et al., 2024b; Zhang et al., 2024; Wang et al., 2024a; Lei et al., 2024) have made significant progress in recovering dynamic scenes along with camera trajectories. These methods typically infer motion masks to separate static regions. To assess practical applicability, we integrate our motion masks into the pipelines and conduct evaluations.

**Datasets and Evaluation.** We evaluate on two datasets: Sintel (Butler et al., 2012) and Bonn (Palazzolo et al., 2019), which are synthetic and real-world datasets, respectively, with ground-truth depth and camera poses. Following prior works (Zhang et al., 2024), we assess the performance from two aspects: 1) Depth estimation, evaluated by Abs Rel, RMSE, and $\delta < 1.25$. 2) Camera pose estimation, evaluated by ATE, RPE (translation), and RPE (rotation).

**Baselines.** We select two state-of-the-art, easy-to-implement methods for evaluation. The first one is DUSt3R (Wang et al., 2024c), which first predicts pixel-aligned pointmaps from input images and then applies global alignment optimization to obtain the final reconstruction. Although designed and trained for static 3D scenes, we test whether incorporating motion masks can improve its performance on dynamic scenes. The second baseline is MonST3R (Zhang et al., 2024), which fine-tunes DUSt3R on dynamic scene data and introduces a set of dynamic scene optimization losses. These losses primarily align the estimated point motion with off-the-shelf optical flow in static regions, using motion masks. Please refer to the supp. for further details.

**Results.** Results are shown in Table 3 and Table 4: 1) Our motion masks improve 4D reconstruction in both depth and camera pose estimation, outperforming MonST3R's motion masks. 2) Even for DUSt3R, designed for static scenes, our motion masks significantly enhance performance in dynamic scenarios.

## 4.4 ABLATION STUDY

We conduct ablation of our key design choices on the DAVIS 2016 dataset, shown in Table 5. For more ablation, please refer to supp. Configurations A1-A4 compare different sampling strategies, including random sampling, grid sampling on the epipolar error map, FPS with a fixed number of points, and our iterative FPS sampling. Results indicate that our iterative FPS achieves the best performance due to its well-distributed coverage and adaptive point selection. Configurations B1-B5 examine the effect of different optical flow time intervals. Results show that using multiple time intervals in both forward and backward directions improves the ability to identify moving regions.

| Config | Intervals | Sampling | $IoU \uparrow$ |
|---|---|---|---|
| A1 | $+2$ | random-10 | 74.3 |
| A2 | $+2$ | grid-10 | 77.6 |
| A3 | $+2$ | FPS-10 | 82.0 |
| A4 | $+2$ | iterative-10 | 84.1 |
| B1 | $\pm 1$ | iterative-10 | 85.4 |
| B2 | $\pm 2$ | iterative-10 | 85.7 |
| B3 | $\pm\{1, 2\}$ | iterative-10 | 87.1 |
| B4 | $+\{1, 2, 4, 8\}$ | iterative-10 | 86.8 |
| **B5 (Ours)** | $\pm\{1, 2, 4, 8\}$ | iterative-10 | **87.4** |

Table 5: **Ablation study** of different optical flow time intervals and different point sampling strategies on DAVIS 2016.

## 4.5 DISCUSSION ON FLAWS IN CURRENT EVALUATION

Although we follow prior works by performing evaluation after Hungarian Matching (HM), we argue that this is unrealistic for real-world applications, as in-the-wild videos lack ground-truth masks for HM. In fact, evaluating without HM is more reasonable, as false positives would otherwise be ignored. To investigate this, we re-evaluate our method and the state-of-the-art approach, FlowP-SAM, without HM in Table 6. The results show a significant performance drop for FlowP-SAM (IoU dropping from 86.1 to 42.23), suggesting that current methods may achieve high true positive rates at the cost of excessive false positives. In contrast, our method produces fewer false positives.

| Method | Eval | $IoU \uparrow$ |
|---|---|---|
| FlowP-SAM | Hungarian | 86.1 |
| E-P (B3) | Hungarian | 87.1 |
| E-P (B5) | Hungarian | 87.3 |
| FlowP-SAM | Non-Hungarian | 42.2 |
| E-P (B3) | Non-Hungarian | 78.6 |
| E-P (B5) | Non-Hungarian | 76.8 |

Table 6: **Discussion** of evaluation under different settings. Results are evaluated on DAVIS 2016, where "E-P" denotes our proposed Epipolar Prompt.

However, we acknowledge that this setting is partly limited by the lack of datasets with all moving-object annotations (see Figure 2). As a result, even real moving objects may be incorrectly classified as false positives due to missing ground-truth labels. Hence, we pose this dilemma to the community and hope a dedicated dataset can be constructed for fair evaluation in the future.

## 5 CONCLUSION

In this paper, we propose *Epipolar Prompt*, a prompt-based motion segmentation method that formulates motion segmentation as a point prompt selection problem. By integrating epipolar geometry cues with the Segment Anything Model (SAM), our approach consists of three key components: epipolar error map calculation, iterative FPS sampling, and heuristic filtering. Experimental results show that *Epipolar Prompt* achieves state-of-the-art performance on both single- and multi-object

benchmarks. Furthermore, the improvements in 4D reconstruction with our motion masks also high-light the practical significance of our method.

**Ethics Statement** We affirm that our work is consistent with the ICLR Code of Ethics. In conducting this research, we considered potential ethical issues such as (i) privacy and data handling, (ii) bias and fairness, (iii) misuse or dual-use risk, and (iv) conflicts of interest. For data that include human-associated content, we apply anonymization / de-identification and obtain appropriate permissions or IRB approval as necessary. We analyze bias across relevant demographic groups and report any limitations. Any sponsors or funding sources are disclosed, and we take responsibility for any potential misuse of the methods.

**Reproducibility Statement** We have aimed to ensure that our experimental results are fully reproducible. All architectural and algorithmic details are described in the main text and Appendix. The code will be released upon acceptance.

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

# A APPENDIX

## A.1 THE USE OF LARGE LANGUAGE MODELS (LLMs)

The research ideas and experimental design of this paper were conceived entirely by the authors without the use of LLMs. During manuscript preparation, we used GPT to assist with grammar checking and refinement of language for clarity and readability.

## A.2 LIMITATIONS AND FUTURE WORK

Despite its effectiveness, our method is inherently a frame-level motion segmentation approach, which cannot capture momentary motionlessness. Additionally, inaccuracies in optical flow estimation can directly impact the results. Moreover, since epipolar geometry reflects only geometric consistency rather than semantic motion, the fitting process is biased toward whichever region dominates the image. In rare cases where a large moving foreground occupies most of the frame, the smaller background region may be incorrectly highlighted as the outlier.

Future work could explore sequence-level motion segmentation by considering temporal consistency or long-range dependencies to enhance robustness and stability. And integrating motion segmentation with reconstruction could improve accuracy by leveraging the complementary nature of the tasks. Moving regions, with higher reconstruction uncertainty, could serve as a cue for refining segmentation, while accurate motion segmentation can help distinguish dynamic and static regions for better reconstruction. In addition, incorporating world knowledge or semantic priors is a promising direction for mitigating the rare failure case described above. Such priors can help disambiguate scenes where static regions do not dominate the image, reducing reliance on purely geometric cues. Furthermore, prompt learning is a promising direction to enhance both accuracy and robustness. We hope our work provides a fruitful basis for future research in this area.

## A.3 RUNTIME ANALYSIS

The average runtime per frame is approximately 0.9 seconds before VLM-based filtering, with most of the time spent on SAM inference. The computation of the epipolar error mask and edge-coverage filtering is highly efficient and has negligible impact on runtime. VLM-based filtering takes an additional 2–3 seconds for enhanced filtering.

## A.4 MORE IMPLEMENTATION DETAILS

**SAM**    We use SAM-ViT-Huge in multi-mask mode, which generates three masks to capture different levels of granularity. This helps prevent partial segmentation issues that may arise in single-mask mode when the point prompt falls within only a part of the object.

**Scoring SAM Masks**    Since SAM's predictions are not always reliable, we apply the same confidence score and stability score filtering as in SAM (Kirillov et al., 2023). The confidence score is directly provided by SAM for each predicted mask. A mask is considered stable if thresholding the probability map at $0.5 - \delta$ and $0.5 + \delta$ results in similar masks.

The stability score is measured by comparing two binary masks derived from the same soft mask but thresholded at different values. A predicted mask (i.e., the binary mask obtained by thresholding logits at 0) is retained only if the IoU between its corresponding $-0.5$ and $+0.5$ thresholded masks is at least 95.0.

**VLM-Based Filtering**    We use Qwen2.5-VL-7B for VLM-based filtering. The prompt used is shown in Figure 8, and example results are presented in Figure 9.

---

**Qwen2.5-VL Prompt**

```
Carefully examine the image and identify any visible objects and static
background elements, and return their names in json format.

Common static background elements include but are not limited to:
rocks, ground, road, buildings, sky, wall, trees, mountains, water,
grass, sea, wall, floor, sand.

Common objects include but are not limited to:
person, car, bicycle, motorcycle, airplane, bus, train, truck, boat,
flamingo, paraglider, sailboard.

Example response: \{

    ``objects": [
        ``car",
        ``train",
    ],

    ``static\_elements": [
        ``road",
        ``water",
    ]

\}

STRICT REQUIREMENT: ENSURE YOUR RESPONSE FOLLOWS THE FORMAT OF THE
EXAMPLE RESPONSE.

REMEMBER: CAREFULLY EXAMINE THE IMAGE AND DO NOT MISS ANY OBJECTS OR
STATIC BACKGROUND ELEMENTS.
```

---

Figure 8: Example of the prompt used in Qwen2.5-VL to filter static background.

## A.5 MORE EXPERIMENTS

### A.5.1 DATA QUALITY

Since SegTrack v2 and FBMS59 are relatively old datasets, some of the videos in these datasets have low quality, with low resolution and even corrupted images (e.g., mosaic effects), which pose challenges for optical flow estimation. Some examples are shown in Figure 10.

### A.5.2 MORE ABLATION STUDY

**Epipolar Error Estimation** Epipolar error estimation is a crucial step in computing the epipolar error map, as both optical flow and the robust estimation of the fundamental matrix directly impact subsequent steps. Under the same optical flow time interval ($+2$) and sampling strategy (FPS-10), Table 7 presents results for different optical flow methods, including RAFT (Teed & Deng, 2020), GMFlow (Xu et al., 2022), and SEA-RAFT (Wang et al., 2024d), as well as robust estimation methods such as MAGSAC (Barath et al., 2019), USAC (Raguram et al., 2012), RANSAC (FIS-CHLER AND, 1981), and LMedS (Rousseeuw, 1984). The results indicate that RAFT combined with LMedS yields the best performance, making it our chosen approach for the main experiments.

**Heuristic Filtering** Since evaluation with Hungarian matching does not account for false positives, we conduct experiments without Hungarian matching to assess the impact of heuristic filtering strategies. Table 8 presents the results, showing that both filtering strategies effectively reduce false positive masks, and their combination further enhances performance. However, since VLM-based

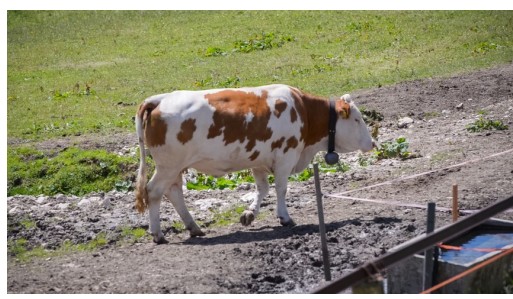 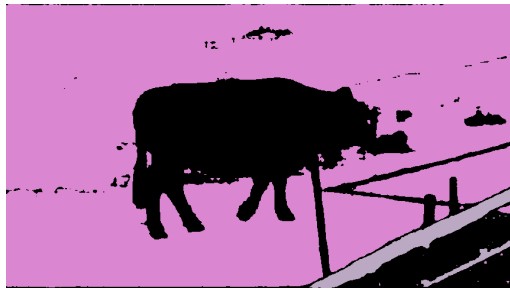

{'objects': ['cow'], 'static_elements': ['grass', 'rocks', 'water', 'fence', 'ground']}

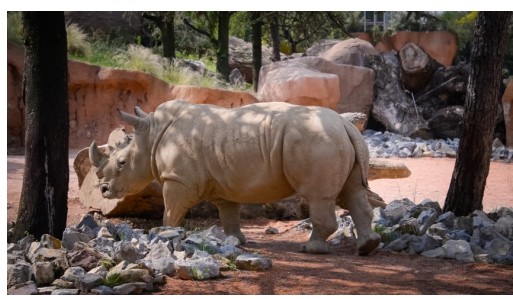 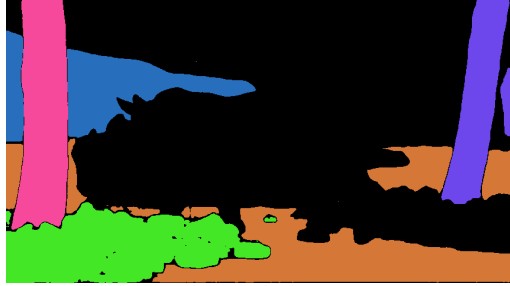

{'objects': ['rhinoceros'], 'static_elements': ['grass', 'trees', 'rocks', 'wall', 'ground']}

Figure 9: Example illustration of VLM outputs and resulting background masks.

| Config | Flow | Rob. Est | $IoU \uparrow$ |
|--------|------|----------|------|
| C1 | RAFT | MAGSAC (Barath et al., 2019) | 71.915 |
| C2 | RAFT | USAC (Raguram et al., 2012) | 77.329 |
| C3 | RAFT | RANSAC (FISCHLER AND, 1981) | 78.469 |
| **C4** | **RAFT** (Teed & Deng, 2020) | **LMedS** (Rousseeuw, 1984) | **82.050** |
| D1 | GMFlow (Xu et al., 2022) | LMedS | 81.582 |
| D2 | SEA-RAFT (Wang et al., 2024d) | LMedS | 79.054 |

Table 7: **Ablation study.** "Flow" for optical flow estimation methods "Rob. Est" for "Robust Estimation". The time interval is +2, and the sampling strategy is FPS-10 for these experiments. Evaluated on DAVIS 2016.

filtering is more time-consuming, for real-time applications requiring higher efficiency, we recommend using only edge coverage filtering, which retains most of the accuracy while significantly reducing computation time.

SegTrack v2                    FBMS59

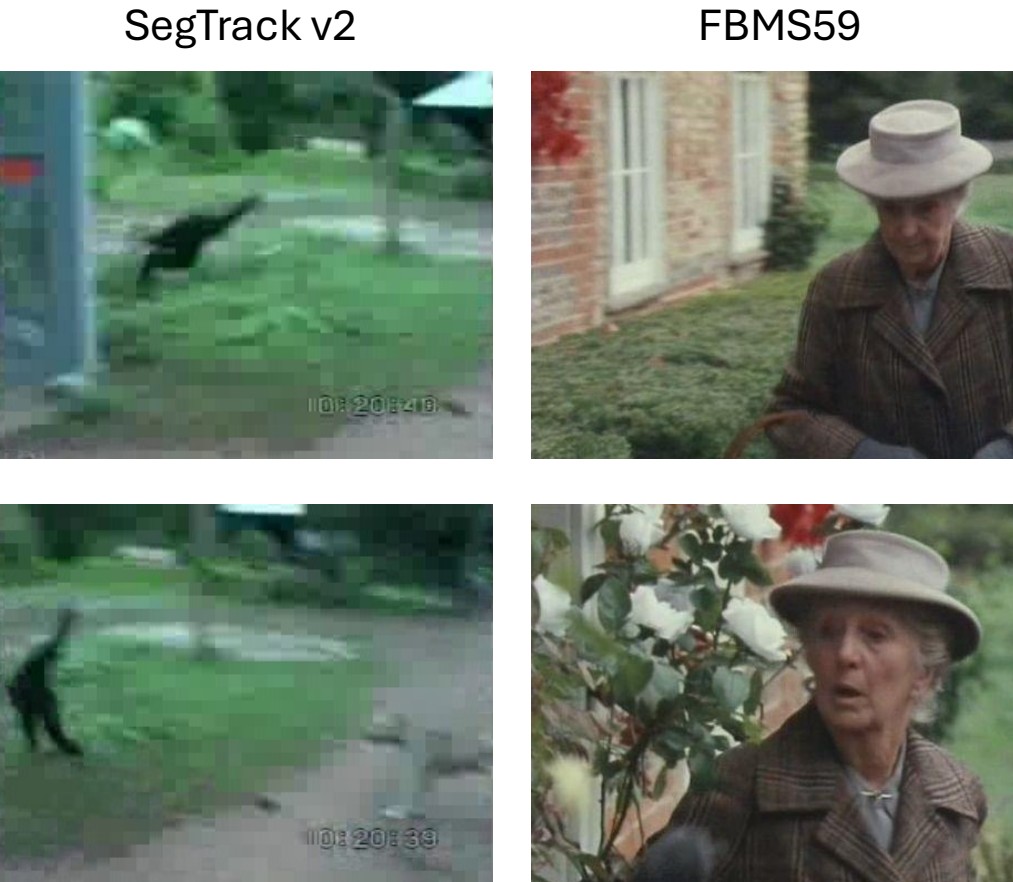

Figure 10: Example illustrations of low-quality videos in SegTrack v2 and FBMS59.

| Method | Eval | $IoU \uparrow$ |
|---|---|---|
| FlowP-SAM | Non-Hungarian | 42.23 |
| E-P (B3) w/o both | Non-Hungarian | 62.979 |
| E-P (B3) w/o EC | Non-Hungarian | 77.538 |
| E-P (B3) w/o VLM | Non-Hungarian | 78.553 |
| **E-P (B3)** | Non-Hungarian | **78.621** |
| E-P (B5) w/o both | Non-Hungarian | 45.188 |
| E-P (B5) w/o EC | Non-Hungarian | 68.00 |
| E-P (B5) w/o VLM | Non-Hungarian | 70.313 |
| **E-P (B5)** | Non-Hungarian | **72.270** |

Table 8: **Ablation study** of heuristic filtering: "E-P" stands for our Epipolar Prompt, "EC" refers to edge coverage filtering, and "VLM" denotes VLM-based filtering. The evaluation is performed on DAVIS 2016.

