# OpenReview forum: "Epipolar Prompt: A Simple Baseline for Motion Segmentation"
_ICLR.cc/2026/Conference — Submitted to ICLR 2026_

### Official Review · Reviewer_eqwm · 2025-10-28

**Soundness:** 3
**Presentation:** 3
**Contribution:** 2
**Rating:** 4
**Confidence:** 4

**Summary:**

The paper proposes Epipolar Prompt, a training-free network for robust motion segmentation between pairwise images. The authors leverage a pretrained optical flow model to predict correspondences, which are then used to estimate the relative camera pose and compute epipolar error maps. The epipolar error is then used as a cue to prompt the foundation segmentation model SAM to achieve precise motion segmentation. The authors show that the proposed pipeline outperforms both supervised and unsupervised approaches on DAVIS2017 and in-the-wild videos. It can also be used with 4D reconstruction models for better camera tracking.

**Strengths:**

- The paper proposes a simple and training-free solution for motion segmentation, which leverages pretrained optical flow and segmentation models SAM.
- The authors use an epipolar-guided error map to prompt the SAM model, providing useful dynamic cues by estimating camera pose and further improving segmentation through the foundation model.
- The authors demonstrate state-of-the-art performance on multiple benchmarks, outperforming both supervised and unsupervised methods.

**Weaknesses:**

- Limited novelty. As the authors mention in the introduction, the idea of using epipolar maps has also been explored in RoMo (Goli et al., 2024) and other previous works. The processing steps are similar: (1) use optical flow to obtain correspondences; (2) use correspondences to recover the fundamental matrix and compute epipolar error. This paper uses the epipolar error map to prompt the SAM model, while RoMo uses the error map to train a classifier.
- Reliability of the epipolar error map in dynamic scenes. The authors propose recovering the fundamental matrix via RANSAC. However, on challenging dynamic scenes (e.g., Sintel), RANSAC often fails to capture static correspondences correctly, which leads to inaccurate camera poses. As a result, the epipolar error map may not provide useful cues for these scenes.
- Comparison to MonST3R masks and runtime. As shown in Table 3, the proposed mask performs similarly to the MonST3R mask, which doesn’t require a foundation segmentation model. Also, as mentioned in the supplementary, the runtime with VLM-based filtering takes about 2–3 seconds per pair, which is slow compared to other methods like the MonST3R mask.

**Questions:**

- It would be helpful if the authors compared their approach with RoMo (Goli et al., 2024) and clarified the specific contributions. In addition, the authors should include a comparison with RoMo in Table 2.
- Since RANSAC can handle only limited amounts of dynamic motion, how robust is the proposed method for predicting motion segmentation if RANSAC fails to obtain the static correspondences?
- In Table 3, the authors show that incorporating the motion segmentation mask improves depth estimation. The authors should further explain how the mask improves depth performance in DUSt3R/MonST3R, as this is not trivial.
- The proposed method relies on pairwise optical flow and camera pose estimation for motion segmentation. How does the method extend to video sequences? Can it produce temporally consistent segmentation of the same object?

---

> ### Author Response · Authors · 2025-12-03
> **Response to Reviewer eqwm**
>
> We thank the reviewer for their helpful feedback. We address your concerns below, where **W** refers to weakness and **Q** refers to questions.
>
> ---
>
> **[W1 & Q1]** Thanks for the comment. We have now added a comparison to RoMo and clarified the distinctions. Although both methods compute an epipolar error map, **the goal, formulation, and expressive power are fundamentally different**. RoMo uses epipolar error solely as a supervised training signal to learn a single binary motion mask, whereas we introduce a new training-free paradigm that uses epipolar cues to prompt SAM and produce multiple object-level motion masks—a capability RoMo does not have.
>
> This conceptual shift is reflected in performance: **our method surpasses RoMo by a large margin** across all benchmarks, despite being a very simple baseline within this new paradigm. The fact that such a lightweight instantiation already exceeds existing approaches demonstrates the strong potential of the paradigm itself. We believe that further exploration along this direction will be highly promising and will substantially benefit the community.
>
> ---
>
>
> **[W2 & Q2]** Thanks for raising this concern.
> We would like to clarify that our method **does not use naive RANSAC**, and instead incorporates several design choices that explicitly mitigate this issue:
>
> 1. *Multi-interval flow aggregation (Sec. 3.1)*:
> By aggregating forward/backward correspondences across multiple time intervals, outliers from dynamic regions are greatly reduced. Our ablations (Table.5) show that this significantly stabilizes the epipolar cues, even on challenging data such as Sintel.
>
> 2. *Robust estimation via LMedS rather than vanilla RANSAC (L193-L201)*:
> We intentionally use LMedS, which is more robust to large fractions of dynamic outliers and does not assume a majority of static correspondences. This improves reliability in scenes where vanilla RANSAC may fail, as ablated in Appendix Table.7.
>
> 3. *Purpose of the epipolar map is guiding prompts—not precise pose recovery*:
> Our paradigm does not require accurate camera poses; it only needs approximate localization of motion inconsistencies, for which the epipolar error remains highly informative even when pose estimation is imperfect.
>
> 4. *Empirical evidence: strong real-world performance despite dynamic scenes*:
> Datasets like DAVIS17 and YouTube-VOS contain severe object motion, yet the epipolar cues remain effective enough to produce state-of-the-art performance. This demonstrates that the estimated F-matrix is sufficiently reliable for our prompting framework, even without perfect camera poses.
>
> ---
>
> **[W3]** The purpose of Table.3 and Table.4 is to demonstrate that our method can also benefit downstream 4D reconstruction tasks, not to propose a new 4D reconstruction method. The results already show that our predicted motion masks consistently outperform those of MonST3R and FlowP-SAM. Existing works such as RoMo also include downstream task experiments (e.g., SfM) to validate the effectiveness of their motion segmentation. Regarding runtime, as described in Appendix L853–L856, the VLM filtering can be disabled in practice for much faster speed (0.6s per-frame) with only a minor performance impact.
>
> ---
>
> **[Q3]** Thank you for the question. DUSt3R/MonST3R optimize a global pointmap by aligning per-frame pointmaps (projected depths) under a static-scene assumption. Dynamic points violate this assumption and introduce inconsistent constraints, which distort the alignment and degrade depth accuracy. Using motion mask to exclude dynamic regions from the alignment loss removes these contradictory signals, allowing the optimizer to recover a cleaner global pointmap and thus better depth estimates.
>
> ---
>
> **[Q4]** Yes. We apply Hungarian matching across frames to associate mask identities across time, and our webpage provides extensive video results demonstrating consistent multi-frame object segmentation, as stated in abstract ("View results at: https://anonymous-for.github.io/ICLR-4426/").

---

### Official Review · Reviewer_Q59N · 2025-10-30

**Soundness:** 3
**Presentation:** 3
**Contribution:** 2
**Rating:** 2
**Confidence:** 4

**Summary:**

This paper proposes Epipolar Prompt, a zero-shot motion segmentation framework that integrates epipolar geometry with the Segment Anything Model (SAM). The key idea is that violations of epipolar geometry in optical flow correspondences can identify moving regions in a scene. These regions are then used to prompt SAM to produce motion segmentation masks without training or fine-tuning.

**Strengths:**

1. The paper smartly combines epipolar geometry—a classical geometric constraint—with the Segment Anything Model (SAM).
2. The method does not require training or fine-tuning, making it lightweight and adaptable.
3. The results show the effectiveness of the proposed method.

**Weaknesses:**

1. Dependence on heuristic thresholds: The approach relies heavily on manually tuned parameters such as the epipolar error threshold and confidence/stability scores. This dependence can make performance sensitive to dataset variations and limit robustness in uncontrolled environments.
2. Limited performance on older benchmarks: Although strong overall, the method performs slightly worse than FlowP-SAM on SegTrack v2 and FBMS59, suggesting that it may struggle with low-quality or complex motion data.
3. Vulnerability to optical flow errors: Since the method builds on optical flow correspondences, inaccuracies from flow estimation (due to blur, occlusion, or lighting changes) can directly affect segmentation quality, producing false motion regions.
4. Engineering-heavy, limited theoretical novelty: The contribution lies mainly in the integration of existing tools (epipolar geometry, SAM, heuristic filtering) rather than a fundamentally new theoretical framework, which may reduce its perceived novelty for top-tier conferences.
5. Although the paper provides quantitative benchmarks and qualitative frame examples, the absence of supplementary video results (or at least a more extensive collection of visual examples in the appendix) makes it difficult to fully assess the temporal consistency and perceptual quality of the motion segmentation. Providing such results would strengthen the empirical evidence and transparency of the work.

**Questions:**

1. The authors mostly follow the evaluation datasets and metrics used in [1]. However, it is unclear why the MOCA dataset [2] was not included in the evaluation. Could the authors explain the reason for excluding this dataset?
2. The combined FlowP-SAM + FlowI-SAM model in [1] demonstrates higher performance than the individual models. It would strengthen the comparison if the authors included the results of FlowP-SAM + FlowI-SAM in Table 2. Since this combined model is likely larger, comparing the number of parameters would further highlight the efficiency advantage of the proposed method.
3. Could the authors also report the model size and inference time to better illustrate the computational efficiency of the proposed approach?
4. How sensitive (or robust) the performance is with respect to the thresholds (confidence, stability, IoU)?


References
[1] Xie, Junyu, et al. "Moving Object Segmentation: All You Need is SAM (and Flow)." Asian Conference on Computer Vision. Singapore: Springer Nature Singapore, 2024.
[2] Lamdouar, Hala, et al. "Betrayed by Motion: Camouflaged Object Discovery via Motion Segmentation." Proceedings of the Asian Conference on Computer Vision, 2020.

---

> ### Author Response · Authors · 2025-12-03
> **Response to Reviewer Q59N (1/2)**
>
> We thank the reviewer for their thoughtful comments. We address your concerns below, where **W** refers to weakness and **Q** refers to questions.
>
> ---
>
> **[W1]** Thanks for the comment. We use a single fixed **set of parameters for all datasets**, and the method performs well without any dataset-specific tuning, indicating good generalization. The thresholds we adopt (e.g., confidence and stability scores) are exactly those used in SAM. And SAM itself even includes more heuristic thresholds than our method (see SAM Appendix B). We reported these details purely for transparency and reproducibility rather than because the method requires delicate tuning.
>
> ---
>
> **[W2]**
> We clarify that the slightly lower numbers on SegTrack v2 and FBMS59 in Table.2 are largely a consequence of the prior questionable evaluation protocol rather than a weakness of the proposed paradigm. As discussed in Sec. 4.5, the Hungarian-matching protocol in Table.2 omits false positives, which disproportionately benefits supervised methods like FlowP-SAM with overflooding false positives to increase matching success rate. To provide a more fair comparison, we additionally report results under the w/o-Hungarian evaluation setting in Table.A (which extends Table.8 in Appendix) below:
>
> **[Table A] (IoU w/o Hungarian Matching)**
> | Method | YTVOS18-m | DAVIS17 | DVIS16 | SegTrackv2 | FBMS59 |
> | :--- | :---: | :---: | :---: | :---: | :---: |
> | Ours | 66.49 | 71.53 | 72.27 | 71.16 | 71.38 |
> | FlowP-SAM | 57.85 | 35.85 | 42.52 | 42.18 | 43.25 |
>
> As shown, under this realistic evaluation protocol, FlowP-SAM performs substantially worse, while our method **outperforms it by a large margin** on all benchmarks, including SegTrack v2 and FBMS59. This demonstrates that the earlier minor gaps on legacy datasets arise from the evaluation artifact rather than limitations of our approach.
>
> Furthermore, FlowP-SAM’s isolated wins in Table 2 are due to data overfitting, as evidenced by its −19 IoU cross-dataset drop reported in Table 1, whereas our training-free formulation generalizes robustly across diverse datasets and real-world scenarios.
>
> ---
>
> **[W3]** Thank you for the comment. As shown in Table 2, **all existing methods rely on optical flow as input**, which demonstrates that optical flow remains a generally reliable cue for motion in most scenarios. Our method further mitigates flow noise by sampling prompt points from the epipolar error map, which is calculated by robust LMedS algorithm, making it more robust to blur, occlusion, and illumination changes.
>
> Moreover, under an evaluation setting that properly counts false positives (w/o Hungarian Matching), our method surpasses the SOTA method FlowP-SAM by a large margin (Table.A). This indicates that the proposed paradigm is highly effective at capturing true motion regions, even when optical flow is imperfect.
>
> ---
>
> **[W4]** Thank you for the thoughtful comment. While our method uses established components, the core contribution is conceptual rather than engineering-based. Inspired by the prompting paradigm in LLMs, we introduce **the first geometric-prompting** paradigm that reframes motion segmentation as epipolar-guided point-prompt selection. This formulation has not appeared in prior work and provides a new way to combine geometric consistency with promptable segmentation models—fundamentally different from existing learning-based pipelines.
>
> This paradigm also exposes a **key limitation of current methods**: supervised models overfit heavily due to scarce and incomplete motion labels. Our training-free geometric formulation overcomes this issue and, despite its simplicity, consistently **outperforms fully trained SOTAs**, showing that the novelty lies in the new problem formulation and the principle it demonstrates.
>
> Furthermore, simplicity does not reduce conceptual significance. Many impactful ICLR papers (e.g., MixUp ’18, FreeMatch ’23) demonstrate that a new idea—even when implemented with simple components—can **meaningfully shift a research direction**. Likewise, the strength of our approach stems from the paradigm itself, not architectural complexity, and it opens a promising new direction for motion segmentation. We believe ICLR is an ideal venue to encourage such paradigm-level advances.
>
> ---
>
> **[W5]** We would like to clarify that we **do provide** extensive video results, as stated in the abstract ("View results at: https://anonymous-for.github.io/ICLR-4426/"). We encourage reviewers to view the project website, which includes rich video evidence demonstrating: (1) visual comparisons against state-of-the-art baselines; (2) why current metrics fail to capture false positives; (3) the annotation deficiencies of existing motion segmentation datasets; and (4) what truly constitutes “motion’’ in motion segmentation.

---

> > ### Author Response · Authors · 2025-12-03
> > **Response to Reviewer Q59N (2/2)**
> >
> > **[Q1]** Thank you for the question. We did not include MOCA because it provides only bounding box annotations rather than segmentation masks like other datasets. In addition, most recent methods, including the follow-up work [r1] by the authors of [1], also no longer evaluate on MOCA. We therefore follow the current evaluation practice.
> >
> > [r1] Appearance-Based Refinement for Object-Centric Motion Segmentation
> >
> > ---
> >
> > **[Q2]** Thank you for the suggestion. We have added the results of FlowP-SAM + FlowI-SAM to Table 2 in the revised version. Please note that the combined FlowP-SAM + FlowI-SAM model offers only a small performance gain while requiring much more number of parameters than ours.
> >
> > ---
> >
> > **[Q3]** As described in Appendix A.4, we adopt the default model sizes of SAM (ViT-H) and Qwen2.5-VL-7B. The runtime is reported in Appendix A.3. The average inference time is about 0.9 s per frame, dominated by SAM inference, while the epipolar-error computation and edge-coverage filtering take negligible time. The VLM-based filtering is optional and adds roughly 2–3 s per frame. Since it offers only a modest performance gain (Table 8) by removing a small number of false positives, it can be disabled in practical usage for faster speed.
> >
> > ---
> >
> > **[Q4]** As described in [W1], we adopt the default thresholds of SAM, and their robustness has already been validated in the original paper (SAM Appendix B).

---

### Official Review · Reviewer_cN3G · 2025-10-31

**Soundness:** 3
**Presentation:** 3
**Contribution:** 2
**Rating:** 4
**Confidence:** 4

**Summary:**

The paper introduces a learning-free approach to moving object segmentation via so-called epipolar prompt. Given a pretrained SAM, VLM, and optical flow model, the method first calculate epipolar error map and detect regions that violates the epipolar constraints (ie, moving part). Then it samples few points from the region, prompts SAM using those points, and gets moving object mask. It iteratively updates the mask while being validated via VLM. The method achieves the best/competitive accuracy on the benchmark.

---

The proposed method sounds great with good accuracy. However, there are concerns on the novelty, paper fit, and overstating of 'no training needed' while using more advantageous setups than others, which can mislead. Thus the recommendation is **4: marginally below the acceptance threshold**. However, any thoughts and justifications regarding these concerns would be appreciated!

**Strengths:**

* **Clarity**

  The paper is written clearly and read really well. It includes necessary technical details to fully understand the method. It further includes in depth analyses of the method, such as ablation study, hyperparameter choices, limitations, justifications on the design choices, etc.

* **Good results**

  The methods achieves the competitive/best accuracy among other methods in the benchmark. In the downstream task (camera pose evaluation and depth estimation), the setup with the proposed methods achieves the best accuracy over other setups using other methods.

**Weaknesses:**

* **Novelty concern / paper fit**

  Though the paper shows good accuracy on the benchmark, I cannot erase an impression that the method is likely a well-composed pipeline of established methods using heuristics (one could interpret it as a contribution of the method though, ie simple composition of off-the-shelf methods beat learning-based algorithms). The idea is neat indeed. However, I am not so sure if there can be any new novel finding or learning representation learned from the paper.

  Call of papers in the ICLR webpages says that it accepts applications in vision, but the paper seems quite at the end of the application side (close to WACV, for example). I was wondering if the paper would fit to the ICLR's interests.


* **Argument on 'no training needed'**

  In Table 2, the paper classifies its method as `No training needed`, which is true. Though the proposed technique itself is `No training needed`, the whole pipeline is based on a more advantageous setup using several off-the-shelf methods including RAFT, SAM, and VLM (with 7M parameters). This argument might mislead others.

  Also, without the usage of VLM, how much does the accuracy drop on each dataset in Table 2? (Table 8 shows only partial results)

  Also in Table 2, some numbers (on SegTrack v2 and FBMS59) underperform FlowP-SAM. It's fine because the method still outperforms on the other benchmarks and downstream tasks as well. However, given its advantageous setup, it weakens the strength of the paper.

**Questions:**

* **Minor**

  In Table 4, the number on Sintel, `DUSt3R w/ FlowP-SAM mask`, I think the RMSE 0.5111 might be a typo.


* **Accuracy of the epipolar map**

  I was wondering if there are any analyses on the accuracy of the epipolar error map and curious how accurate/reliable it is. (it may not be necessary but good to evaluate the accuracy on a dataset with GT camera pose)

* **Extreme scenario**

  I was also wondering if there is any limitation coming from the epipolar map. For example, it's a rare case that other methods can fail as well, but let's assume an image pair where a foreground object dominate most of the image part, and the epipolar error map highlights the background region. Then how will the method behave?

* **What's the runtime of the method?**

  Especially it's curious how much time does each stage take.

---

> ### Author Response · Authors · 2025-12-03
> **Response to Reviewer cN3G**
>
> We thank the reviewer for their constructive feedback. We address your concerns below, where **W** refers to weakness and **Q** refers to questions.
>
> ---
>
> **[W1]**
> We thank the reviewer for the thoughtful comments. While our method uses established components, we respectfully argue that the key contribution is **conceptual**. Motivated by the success of *prompting in LLMs*, we are **the first** to introduce a new **geometric-prompting paradigm** that reframes motion segmentation as epipolar-guided point-prompt selection. This perspective is novel and reveals a new finding: epipolar cues naturally serve as effective prompts for motion masks, enabling strong generalization.
>
> This new paradigm overcomes a **long-standing limitation** of prior learning-based methods: severe overfitting caused by scarce and incomplete motion labels. Our simple yet principled formulation achieves strong generalization and consistently surpasses supervised and unsupervised baselines, underscoring the scientific value of the paradigm itself.
>
> We also believe that simplicity should not diminish conceptual innovation. Many influential ICLR works (e.g., MixUp ’18, FreeMatch ’23) show that **simple mechanisms can meaningfully shift a research direction** when the core idea is new. Our results demonstrate that this paradigm—not engineering complexity—is what brings **surprising performance gains**.
>
> Additionally, our work provides an honest view of a problematic trend: current evaluation protocols foster overfitting and mask the lack of true geometric reasoning in existing models. Our formulation and rectified metrics help expose this issue and open a new research direction that we detail in the appendix. We believe ICLR is an ideal venue to encourage such paradigm-level advances.
>
> ---
>
>
> **[W2]** Thanks for the helpful comments. We will rephrase "no training needed" as "training-free," which is adopted by many published papers [1,2,3].
>
> [1] BoxDiff: Text-to-Image Synthesis with **Training-Free** Box-Constrained Diffusion [ICCV 2023]
> [2] FreeControl: **Training-Free** Spatial Control of Any Text-to-Image Diffusion Model with Any Condition (CVPR 2024)
> [3] **Training-free** image manipulation localization using diffusion models [AAAI 2025]
>
>
> We clarify that Table 2 follows the prior evaluation protocol, where predicted masks are first Hungarian-matched to GT masks **before** IoU computation. However, as discussed in Sec. 4.5, this protocol problematically overlooks false positives, and therefore does not reflect real usage. To provide a fairer comparison, we also report results under the w/o-Hungarian evaluation setting in Table.A (which extends Table.8 in Appendix) below:
>
> **[Table A] (IoU w/o Hungarian Matching)**
> | Method | YTVOS18-m | DAVIS17 | DVIS16 | SegTrackv2 | FBMS59 |
> | :--- | :---: | :---: | :---: | :---: | :---: |
> | Ours w/o VLM | 63.75 | 68.85 | 70.31 | 69.32 | 70.58 |
> | Ours w/ VLM | 66.49 | 71.53 | 72.27 | 71.16 | 71.38 |
> | FlowP-SAM | 57.85 | 35.85 | 42.52 | 42.18 | 43.25 |
>
> As shown, removing VLM only moderately affects our method, whereas FlowP-SAM performs substantially worse under this more realistic evaluation protocol. Our method clearly outperforms FlowP-SAM by a large margin across all benchmarks, further validating that the strength of our approach comes from the proposed paradigm rather than any evaluation artifact.
>
> Moreover, FlowP-SAM’s isolated wins in Table 2 are due to **data overfitting**, as demonstrated in Table.1.
>
> ---
>
> **[Q1]** Thanks for the kind reminder. We have updated it in the revised version.
>
> ---
>
> **[Q2]** Thank you for the suggestion. We have added video results of the epipolar error maps on our anonymous webpage (https://anonymous-for.github.io/ICLR-4426/), allowing direct inspection of their reliability.
>
> ---
>
> **[Q3]** We thank the reviewer for the thoughtful question. Our method assumes that most of the majority of the scene is static, which matches most real-world video scenarios. Since epipolar geometry itself does not distinguish "moving" from "static", actually the error map is dominated by the larger region, and in the rare case where a large moving foreground occupies most of the image, the background may be incorrectly highlighted and other methods also struggle to handle this challenging case. We have added this limitation to the revised manuscript and plan to explore incorporating world knowledge or semantic priors to mitigate such cases.
>
> ---
>
> **[Q4]** The runtime is reported in Appendix A.3. The average runtime per frame is about 0.9 s, dominated by SAM inference; both the epipolar-error computation and the edge-coverage filtering are negligible. The VLM-based filtering is optional and adds roughly 2–3 s per frame. As it provides only a modest performance gain (Table 8) by removing a small number of false positives, it can be disabled in practical usage for faster speed.

---

### Meta-Review · Area_Chair_XFtv · 2026-01-13

**Summary:**

This paper proposes a simple epipolar-based prompting method which uses epipolar error maps along with a prompt selection scheme to be used with models such as SAM. Reviewers had a range of shared concerns. Reviewer cN3G had concerns regarding the novelty and specifically heuristic nature of the method lacking principled contributions. They also raised a number of specific questions regarding the results/tables, including lower performance of the method despite having the advantage of models such as SAM (also mentioned). Further, the reviewer mentioned need for analysis of the epipolar maps themselves to verify the claims. Reviewer Q59N shared concerns regarding the heuristic algorithm, limited performance in some cases, and further mentioned some missing datasets (MOCA). Finally, Reviewer eqwm had similar shared concerns and also mentioned that the proposed method performs similarly to other methods that are significantly less compute since they do not require SAM/foundation models.

**Reviewer Concerns:**

While the reviewers addressed concerns such as hyper-parameters (noting the use of a single set across datasets) and also clarified that the evaluation protocol ignores false positives (due to the Hungarian matching), explaining some of the underperformance. However, many of the concerns above are not resolved; the paper indeed proposes a simple heuristic method, and while simplicity is not necessarily a reason for rejection, the paper does not provide a strong performance-computation tradeoff.

**Reviewer Scores:**

Overall, while some reviewers may raise their score slightly due to the partially addressed concerns mentioned above. As a result, it is unlikely that the reviewers would significantly increase their scores.

---

### Decision · Program_Chairs · 2026-01-26

Reject